# Propofol-based total intravenous anesthesia is associated with better survival than desflurane anesthesia in robot-assisted radical prostatectomy

**Hou-Chuan Lai[1], Meei-Shyuan Lee[2], Kuen-Tze Lin[3], Yi-Hsuan Huang[1], Jen-Yin Chen[4,5], Yao-Tsung Lin[4], Kuo-Chuan Hung[4], Zhi-Fu Wu[1,4]***

1 Department of Anesthesiology, Tri-Service General Hospital and National Defense Medical Center, Taipei, Taiwan, Republic of China, 2 School of Public Health, National Defense Medical Center, Taipei, Taiwan, Republic of China, 3 Department of of Radiation Oncology, Tri-Service General Hospital and National Defense Medical Center, Taipei, Taiwan, Republic of China, 4 Department of Anesthesiology, Chi Mei Medical Center, Tainan City, Taiwan, Republic of China, 5 Department of the Senior Citizen Service Management, Chia Nan University of Pharmacy and Science, Tainan City, Taiwan, Republic of China

* aneswu@gmail.com

**Data Availability Statement:** All relevant data are within the manuscript.

## Abstract

### Background

Previous researches have shown that anesthetic techniques may influence the patients' outcomes after cancer surgery. Here, we studied the relationship between the type of anesthetic techniques and patients' outcomes following elective robot-assisted radical prostatectomy.

### Methods

This was a retrospective cohort study of patients who received elective, robot-assisted radical prostatectomy between January 2008 and December 2018. Patients were grouped according to the anesthesia they received, namely desflurane or propofol. A Kaplan–Meier analysis was conducted, and survival curves were presented from the date of surgery to death. Univariable and multivariable Cox regression models were used to compare hazard ratios for death after propensity matching. Subgroup analyses were performed for tumor-node-metastasis stage and disease progression. The primary outcome was overall survival, and the secondary outcome was postoperative biochemical recurrence.

### Results

A total of 365 patients (24 deaths, 7.0%) under desflurane anesthesia, and 266 patients (2 deaths, 1.0%) under propofol anesthesia were included. The all-cause mortality rate was significantly lower in the propofol anesthesia than in the desflurane anesthesia during follow-up ($P = 0.001$). Two hundred sixty-four patients remained in each group after propensity matching. The propofol anesthesia was associated with improved overall survival (hazard ratio, 0.11; 95% confidence interval, 0.03–0.48; $P = 0.003$) in the matched analysis.

**Funding:** The author(s) received no specific funding for this work.

**Competing interests:** The authors have declared that no competing interests exist.

**Abbreviations:** RARP, robot-assisted radical prostatectomy; VAs, volatile anesthetics; BCR, biochemical recurrence; ASA, American Society of Anesthesiology; TNM, tumor–node–metastasis; CCI, Charlson comorbidity index; METs, metabolic equivalents; SD, standard deviation; PS, propensity score; HR, hazard ratio; CI, confidence interval; HIF, hypoxia-inducible factor.

Subgroup analyses showed that patients under propofol anesthesia had less postoperative biochemical recurrence than those under desflurane (hazard ratio, 0.20; 95% confidence interval, 0.05–0.91; $P$ = 0.038) in the matched analysis.

## Conclusions

Propofol anesthesia was associated with improved overall survival in robot-assisted radical prostatectomy compared with desflurane anesthesia. In addition, patients under propofol anesthesia had less postoperative biochemical recurrence.

## Introduction

Prostate cancer is one of the most common malignancy in men, and it is a major cause of morbidity and kills approximately 27,000 people per year in the United States. [1] Recently, the incidence of prostate cancer is increasing in most countries, including Asia. [2] Although there are various treatment options for prostate cancer, radical prostatectomy is recommended for localized prostate cancer patients with a life expectancy > 10 years as a first-line treatment. [2] Robot-assisted radical prostatectomy (RARP) has been widely adopted as a standard procedure for clinically localized prostate cancer worldwide due to less blood loss, lower blood transfusion rate, and less hospitalization stay compared with open radical prostatectomy. [2] Unfortunately, recurrence of prostate cancer after surgery increases postoperative morbidity and mortality. [1] Surgical intervention itself may result in neuroendocrine and metabolic changes which may impair cell-mediated immunity and activation of circulating tumor cell implantation. [3] This potential combination of impaired immune responses and circulating cancer cell seeding enhances the susceptibility of patients undergoing cancer surgery to the development of postoperative recurrence or metastasis, and is associated with poor prognosis.

Recently, the potential role of anesthetic techniques in the process of postoperative recurrence or metastasis formation has attracted attention. [3] Experimental data showed that different anesthetics might affect the immune system in different paths. [4–9] Research has shown that volatile anesthetics (VAs) are pro-inflammatory and might affect immune processes, which might increase the incidence of postoperative recurrence or metastasis. [8–12] However, propofol seemed to reduce tumor proliferation, invasion, and migration and then to decrease the risk of recurrence or metastasis in humans and mice. [6,11–14]

Until now, very few studies have compared the effects of the use of desflurane versus propofol anesthesia on patient outcomes after RARP. We hypothesized that patients under desflurane anesthesia may have poorer overall survival/postoperative biochemical recurrence (primary/secondary hypothesis) than patients under propofol anesthesia as our previous colon cancer and hepatocellular carcinoma studies. [15,16] Thus, we conducted a retrospective cohort study to inspect whether the type of anesthesia, desflurane versus propofol was associated with patient survival and postoperative biochemical recurrence (BCR) following RARP.

## Methods

### Study design and setting

This retrospective cohort study was performed at the Tri-Service General Hospital, Taipei, Taiwan, Republic of China.

## Participants and data sources

The ethics committee of the Tri-Service General Hospital approved this retrospective study and waived the need for informed consent (TSGHIRB No: 1-108-05-119). The information was retrieved from the electronic database and medical records of TSGH. From January 2008 and December 2018, 631 cases with an American Society of Anesthesiologists (ASA) score of II–III who had received elective RARP for tumor-node-metastasis (TNM) of stage I–IV prostate cancer under propofol anesthesia (n = 266) or desflurane anesthesia (n = 365) were eligible for analysis. The type of anesthesia was decided by the anesthesiologist's personal preference. The exclusion criteria were propofol anesthesia combined with VAs or regional analgesia, desflurane anesthesia combined with regional analgesia, incomplete data, age < 20 years. And then, 26 cases were excluded (Fig 1).

No medication was used before the anesthesia induction. Standard monitoring systems, including electrocardiography (lead II), noninvasive blood pressure testing, pulse oximetry, end-tidal carbon dioxide measurement, and direct radial arterial blood pressure were performed in each case. Anesthesia was induced using fentanyl, propofol, and cisatracurium or rocuronium in all cases.

Anesthesia was maintained with target-controlled infusion (Fresenius Orchestra Primea; Fresenius Kabi AG, Bad Homburg, Germany) using propofol at an effect-site concentration of 3–4 μg/mL in $FiO_2$ of 100% oxygen at a flow rate of 300 mL/min in the propofol group. The desflurane vaporizer was set between 4% and 10% in 100% oxygen at a flow of 300 mL/min in a closed breathing system in the desflurane group. Repetitive bolus injections of fentanyl and cisatracurium or rocuronium were used as needed during surgery. [15,16] Desflurane or maintenance of the effect-site concentration with target-controlled infusion using propofol was adjusted downward and upward by 0.5–2% or 0.2–0.5 μg/mL, respectively, when needed based on the hemodynamics. The level of end-tidal carbon dioxide was kept at 35–45 mmHg by

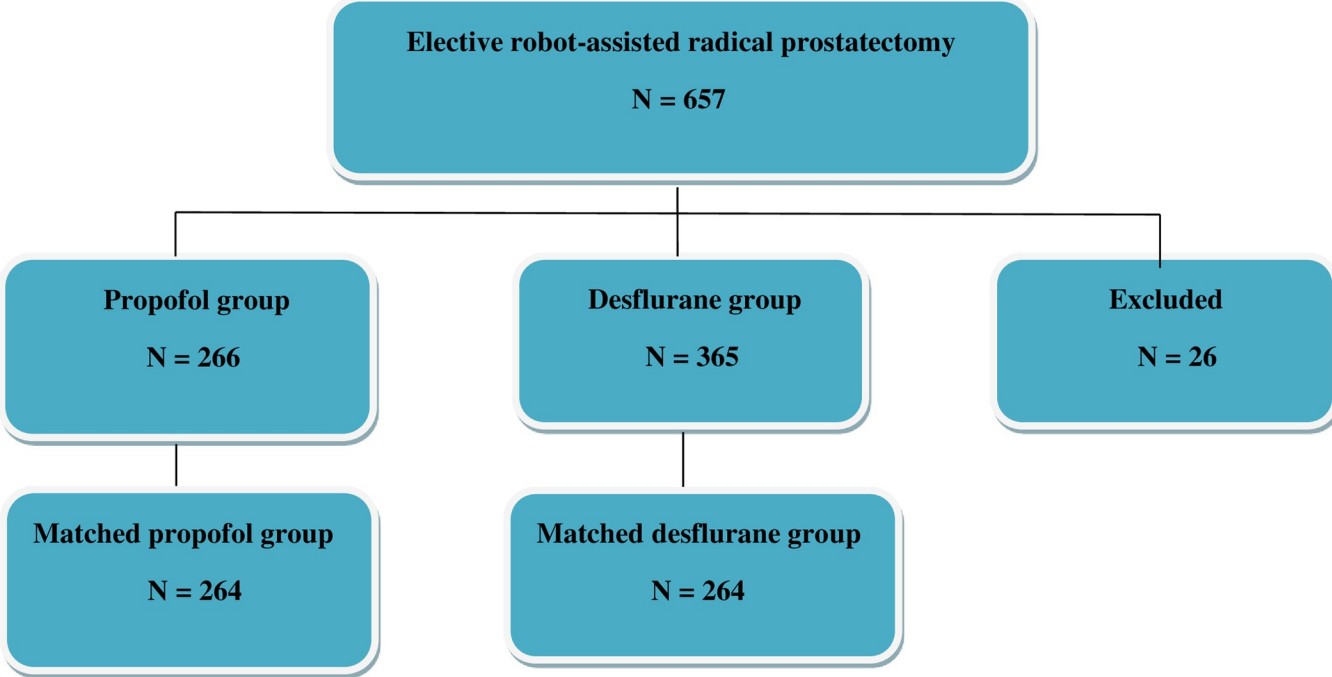

**Fig 1. Flow diagram detailing the selection of patients included in the retrospective analysis.** 26 patients were excluded due to combined propofol anesthesia with inhalation anesthesia or regional analgesia, desflurane anesthesia combined with regional analgesia, incomplete data, age < 20 years.

adjusting the ventilation rate with volume control model with tidal volume 6–8 mg/kg and kept a maximum airway pressure $< 30$ cm $H_2O$. After surgery, all cases were transferred to the postanesthesia care unit for subsequent care.

## Variables

We retrospectively gathered the following patient data: anesthetic technique; time since the earliest included patient, which served as a surrogate of the calendar year; calendar period; sex; age at the time of surgery; and preoperative serum prostate specific antigen (PSA) values. For preoperative PSA levels, patients were grouped according to whether their PSA levels were $> 10$ or $\leq 10$ ng/mL, because a PSA level $\leq 10$ ng/mL was associated with better prognosis in prostate cancer. [17] We used the Charlson Comorbidity Index (CCI) to predict the 10-year survival in patients with multiple comorbidities. The preoperative functional capacity was assessed in metabolic equivalents (METs). Because the cardiac and long-term risks increase in patients with a functional capacity of $< 4$ METs during most normal daily activities, [18] and patients were grouped according to whether the value was $\geq 4$ METs or $< 4$ METs. We also used the Clavien–Dindo classification, scaled from 0 (no complication) to V (most complications), to grade surgical complications. Other data included the ASA physical status score (ranging from I, indicating the lowest morbidity, to V, indicating the highest morbidity); TNM stage of the primary tumor; histological grade of the tumor (low risk: Gleason grade $\leq 6$; medium risk: Gleason = 7; high risk: Gleason $> 7$) [17]; tumor size; intraoperative blood transfusion; postoperative chemotherapy; postoperative radiation therapy; postoperative androgen deprivation therapy; presence of postoperative BCR (defined as two consecutive rises in serum PSA levels $\geq 0.2$ ng/mL at any time postoperatively) [19]; and presence of postoperative metastases. Because these variables have been shown or posited to affect patient outcomes, they were chosen as potential covariates.

## Statistical methods

The primary end point was overall survival, which was compared between the propofol and desflurane groups. The survival time was defined as the interval between the date of surgery and the date of death or July 05, 2019, for those who were censored. All data are presented as mean ± standard deviation (SD) or number (percentage).

Mortality rates and patient characteristics were compared between the groups treated with the different anesthetics using Student's $t$ test, or the chi-square test. The survival according to the anesthetic technique was depicted visually in a Kaplan–Meier survival curve. The association between the type of anesthesia (propofol or desflurane) and survival was analyzed by the Cox proportional-hazards model with and without adjustment for the abovementioned variables as well as surgeons (n = 9). Though no significant interactions with the two anesthetic techniques (propofol or desflurane) were found, due to more than 80% patients with TNM II +III, we also performed subgroup analyses for TNM stage as well as postoperative BCR.

Propensity score (PS) matching with IBM SPSS Statistics 22.0 was used to select for the most similar PSs for preoperative variables (with calipers set at 0.2 SD of the logit of the PS) across each anesthesia: propofol or desflurane in a 1:1 ratio, to make sure the comparability between propofol and desflurane anesthesia before the surgery. Two-tailed $P$-values less than 0.05 were considered statistically significant.

## Results

The patients' and treatment characteristics were shown in Table 1. Desflurane anesthesia had longer time since the earliest included patient compared with propofol anesthesia (5.3 ± 3.0 *vs*

**Table 1. Patients' and treatment characteristics for overall group and matched group after propensity scoring.**

| Variables | Overall Patients | | | Matched Patients | | | |
|---|---|---|---|---|---|---|---|
| | Propofol (n = 266) | Desflurane (n = 365) | p value | Propofol (n = 264) | Desflurane (n = 264) | p value | SMD |
| Time since the earliest included patient (years), Mean (SD) | 4.8 (2.4) | 5.3 (3.0) | 0.014 | 4.8 (2.4) | 4.3 (2.5) | 0.045 | 0.177 |
| Calendar period, n (%) | | | <0.001 | | | 0.080 | 0.133 |
| 2009–12 | 92 (35) | 125 (34) | | 92 (35) | 116 (44) | | |
| 2013–15 | 110 (41) | 97 (27) | | 110 (42) | 89 (34) | | |
| 2016–18 | 64 (24) | 143 (39) | | 62 (24) | 59 (22) | | |
| Age (yr), mean (SD) | 65.3 (6.5) | 66.4 (7.2) | 0.066 | 65.3 (6.5) | 66.3 (7.4) | 0.118 | 0.136 |
| PSA | | | 0.604 | | | 0.254 | 0.119 |
| ≤ 10 | 143 (54) | 205 (56) | | 141 (53) | 155 (59) | | |
| > 10 | 123 (46) | 160 (44) | | 123 (47) | 109 (41) | | |
| Charlson comorbidity index, mean (SD) | 4.1 (0.70) | 4.2 (0.78) | 0.164 | 4.1 (0.70) | 4.2 (0.79) | 0.242 | 0.108 |
| Functional status, n (%) | | | 0.232 | | | 0.549 | 0.131 |
| < 4 METs | 22 (8) | 42 (12) | | 22 (8) | 27 (10) | | |
| ≥ 4 METs | 244 (92) | 323 (89) | | 242 (92) | 234 (90) | | |
| ASA, n (%) | | | 0.232 | | | 0.549 | 0.131 |
| Lower risk | 244 (92) | 323 (89) | | 242 (92) | 234 (90) | | |
| Higher risk | 22 (8) | 42 (12) | | 22 (8) | 27 (10) | | |
| TNM stage of primary tumor, n (%) | | | 0.563 | | | 0.045 | 0.173 |
| I | 44 (17) | 65 (18) | | 43 (16) | 51 (19) | | |
| II | 197 (75) | 274 (75) | | 196 (74) | 202 (77) | | |
| III | 25 (9) | 26 (7) | | 25 (10) | 11 (4) | | |
| Tumor grade (Gleason grade) | | | 0.145 | | | 0.178 | 0.011 |
| Low risk (Gleason grade ≤ 6) | 36 (14) | 45 (12) | | 34 (13) | 24 (9) | | |
| Medium risk (Gleason = 7) | 101 (38) | 167 (46) | | 101 (38) | 119 (45) | | |
| High risk (Gleason > 7) | 129 (49) | 153 (42) | | 129 (49) | 121 (46) | | |
| Tumor size (cm), mean (SD) | 1.6 (0.89) | 1.5 (0.83) | 0.169 | 1.6 (0.89) | 1.6 (0.87) | 0.419 | N/A |
| Intraoperative blood transfusion, n (%) | 10 (4) | 8 (2) | 0.354 | 10 (4) | 6 (2) | 0.446 | N/A |
| Grade of surgical complications, n (%) | | | 0.975 | | | 1.000 | N/A |
| 0 | 257 (97) | 354 (97) | | 255 (97) | 256 (97) | | |
| I + II | 9 (3) | 11 (3) | | 9 (3) | 8 (3) | | |
| Postoperative radiation therapy, yes, n (%) | 32 (12) | 50 (14) | 0.620 | 32 (12) | 34 (13) | 0.895 | N/A |
| Postoperative ADT, yes, n (%) | 42 (16) | 55 (15) | 0.892 | 42 (16) | 46 (17) | 0.726 | N/A |
| Postoperative biochemical recurrence, n (%) | 2 (1) | 13 (4) | 0.043 | 2 (1) | 12 (5) | 0.015 | N/A |
| All-cause mortality, n (%) | 2 (1) | 24 (7) | 0.001 | 2 (1) | 21 (8) | <0.001 | N/A |
| Cancer mortality, n (%) | 0 (0) | 5 (1) | 0.073 | 0 (0) | 4 (2) | 0.055 | N/A |

PSA: prostate specific antigen; METs: Metabolic equivalents; ASA: American Society of Anesthesiologists; TNM: tumor-node-metastasis; ADT: androgen deprivation therapy; SMD: standardized mean differences; N/A: not applicable.

4.8 ± 2.4 years; $P$ = 0.014). The calendar period was significantly different between the two anesthetic techniques ($P$ < 0.001). Age, CCI, preoperative functional status, ASA score, TNM stage of the prostate cancer, preoperative PSA level, tumor size, histological grade of the prostate cancer, grade of surgical complications, need for intraoperative blood transfusion, and the use of postoperative radiation therapy or androgen deprivation therapy were not significantly different between the two anesthetic techniques (Table 1). In addition, no patient was given postoperative chemotherapy.

Table 1 also showed that all-cause mortality rate was significantly lower in the propofol anesthesia (1.0%) than in the desflurane anesthesia (7.0%) during follow-up ($P$ = 0.001). However, the cancer-specific mortality rate was insignificantly different between the two anesthetic techniques during follow-up ($P$ = 0.073). A lower percentage of patients in the propofol anesthesia (1.0%) exhibited postoperative BCR compared with the desflurane anesthesia (4.0%; $P$ = 0.043). There was no postoperative metastasis in the two groups. Kaplan–Meier survival curves for the two anesthetic techniques are shown in Fig 2A.

The overall mortality risk associated with the use of propofol and desflurane during prostate cancer surgery was reported in Table 2. Overall survival from the date of surgery grouped according to the anesthetic technique and other variables were compared individually in a univariable Cox model and subsequently in a multivariable Cox regression model (Table 2). Patients with propofol anesthesia exhibited overall survival compared to those with desflurane anesthesia (overall survival 99.0% versus 93.0%, respectively; the crude hazard ratio (HR) was 0.11 (95% confidence interval (CI), 0.03–0.47; $P$ = 0.003). This finding did not change substantially in the multivariable analyses after adjustment for those significant variables in the univariable analyses and 9 surgeons (HR, 0.12; 95% CI, 0.03–0.54; $P$ = 0.006). After the multivariable analysis, a higher preoperative PSA level was another variable that was identified that significantly increased the mortality risk (Table 2). Kaplan–Meier survival curves for the two anesthetic techniques after PS matching are shown in Fig 2B.

We used the PS from the logistic regression to adjust the baseline characteristics and the choice of therapy between the two anesthetic techniques due to the significant differences in baseline characteristics between the two anesthetic techniques. Two hundred sixty-four pairs were formed after matching (Table 1). Patient characteristics and prognostic factors of prostate cancer were insignificantly different between the matched groups.

## Subgroup analyses for TNM stage and disease progression

In the nonstratified analysis, patients with propofol anesthesia showed better survival than those with desflurane; the crude HR was 0.11 (95% CI, 0.03–0.47; $P$ = 0.003), the PS-adjusted HR was 0.12 (95% CI, 0.03–0.49; $P$ = 0.004), and the PS-matched HR was 0.11 (95% CI, 0.03–0.48; $P$ = 0.003) (Table 3).

Although the PS matching showed that propofol anesthesia provided better outcomes in TNM II+III patients (HR, 0.13; 95% CI, 0.03–0.57; $P$ = 0.007), there was no significant interaction between the type of anesthesia and TNM stage ($P$ = 0.926) (Table 3).

Patients with propofol anesthesia had less postoperative BCR than those with desflurane; the crude HR was 0.20 (95% CI, 0.05–0.91; $P$ = 0.037), the PS-adjusted HR was 0.20 (95% CI, 0.04–0.88; P = 0.033), and the PS-matched HR was 0.20 (95% CI, 0.05–0.91; $P$ = 0.038).

In summary, patients with desflurane anesthesia had more significant disease progression (such as postoperative BCR) than those with propofol anesthesia.

## Discussion

The major findings in the present study are that propofol anesthesia in RARP improves survival and reduces postoperative BCR compared with desflurane anesthesia. The results were consistent with our previous reports demonstrating that propofol anesthesia was related to better survival and a lower incidence of postoperative recurrence or metastasis compared with desflurane anesthesia in colon cancer, hepatocellular carcinoma, and intrahepatic cholangiocarcinoma surgery. [15,16,20,21] By contrast, there were retrospective studies reporting insignificant differences in overall survival between the use propofol and VAs in surgery for breast cancer, lung cancer. [11,22,23] To our best knowledge, there are very few researches of the

**A**

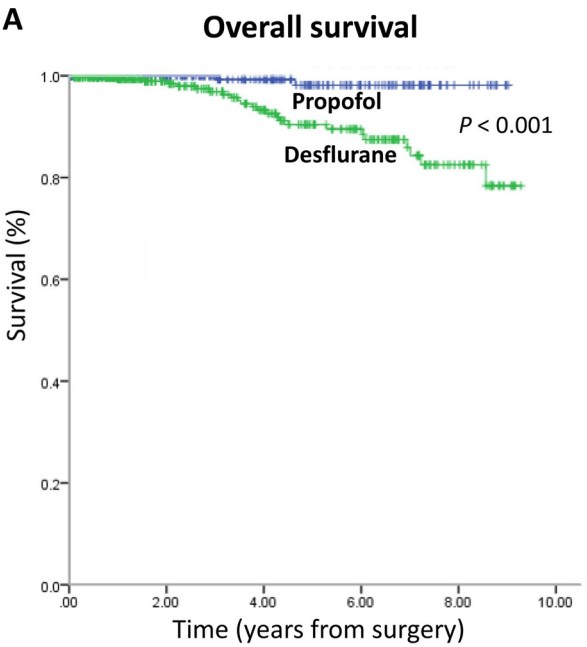

**Overall survival**

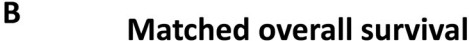

Time (years from surgery)

**B**

## Matched overall survival

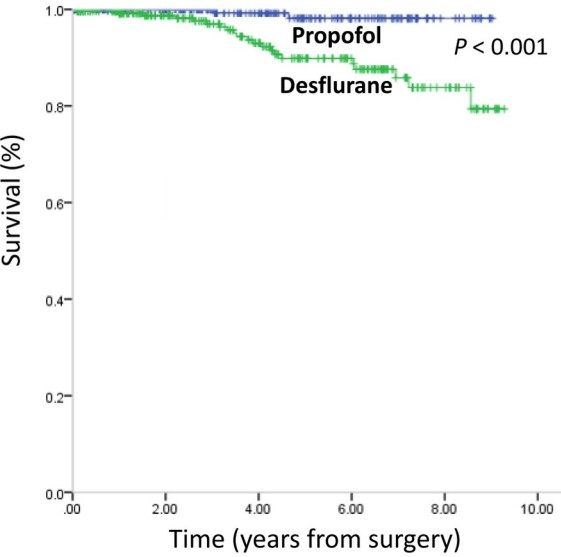

Time (years from surgery)

**Fig 2.** (A) Overall survival curves from the date of surgery by anesthesia type. (B) Overall survival curves from the date of surgery by anesthesia type after propensity score matching.

effects of the propofol and VAs in patients undergoing prostate cancer surgery, and further investigations are needed.

In the current study, we also showed that the overall postoperative BCR was 2.4%, and the postoperative BCR in the matched propofol group (1.0%) was less than in the matched desflurane group (5.0%) after RARP ($P = 0.015$). The result was consistent with Coughlin et al., who reported that postoperative BCR of prostate cancer occurs in 2.5% (4/157) of patients at a

**Table 2. Cox proportional hazards regression for mortality: Univariable and multivariable models for overall patients.**

| | Univariable | | Multivariable | |
|---|---|---|---|---|
| Variables | HR (95% CI) | *P* value | HR (95% CI) | *P* value |
| Anesthesia, Propofol (ref: Desflurane) | 0.11 (0.03–0.47) | 0.003 | 0.12 (0.03–0.54) | 0.006 |
| Time since the earliest op (yr) | 1.24 (0.96–1.60) | 0.095 | | |
| Age (yr) | 1.13 (1.06–1.22) | 0.001 | 1.05 (0.87–1.28) | 0.604 |
| PSA >10 (ref: ≤ 10) | 5.50 (2.07–14.6) | 0.001 | 4.89 (1.74–13.7) | 0.003 |
| Charlson comorbidity index | 2.81 (1.50–5.25) | 0.001 | 1.54 (0.31–7.62) | 0.600 |
| Functional status, ≥4 METs (ref: <4 METs) | 0.27 (0.10–0.72_ | 0.009 | | |
| ASA III, (ref: II) | 3.74 (1.40–10.0) | 0.009 | 0.77 (0.17–3.53) | 0.739 |
| TNM stage of primary tumor (ref: I) | | | | |
| II | 1.28 (0.38–4.31) | 0.690 | | |
| III | 1.64 (0.27–9.86) | 0.587 | | |
| Tumor grade (ref: low+ medium risk) | | | | |
| High risk | 0.92 (0.41–2.06) | 0.838 | | |
| Tumor size (cm) | 1.10 (0.79–1.54) | 0.572 | | |
| Intraoperative blood transfusion (ref: no) | 5.12 (1.75–14.9) | 0.003 | 1.53 (0.28–8.20) | 0.623 |
| Grade of surgical complications (ref: 0) | | | | |
| I+II | 5.91 (2.21–15.8) | <0.001 | 2.14 (0.40–11.4) | 0.374 |
| Postoperative radiation therapy (ref: no) | 1.36 (0.54–3.42) | 0.510 | | |
| Postoperative ADT (ref: no) | 2.42 (1.08–5.41) | 0.032 | 1.58 (0.65–3.89) | 0.315 |
| Postoperative biochemical recurrence (ref: no) | 8.84 (3.31–23.6) | <0.001 | | |

Adjusted-HRs were adjusted by those variables were significant in the univariable analyses and surgeons (n = 9). Functional status was excluded from the multivariable due to it was highly correlated with ASA. Postoperative biochemical recurrence was another outcome, therefore, it was also not included. PSA: prostate specific antigen; METs: Metabolic equivalents; ASA: American Society of Anesthesiologists; TNM: tumor-node-metastasis; ADT: androgen deprivation therapy.

median follow up of 24 months after RARP. [24] In this study, we found that a higher preoperative PSA level was associated with poor survival after prostate cancer surgery, as have been observed previously. [17,25] However, previous data indicated that considerable number of men had prostate cancer despite being within normal PSA range. Importantly, many of these patients were later found to have high grade histology. [26] Therefore, further investigation is necessary for the relationship between PSA or other biomarkers and prognosis of prostate cancer.

**Table 3. Subgroup analyses for TNM stage and disease progression.**

| Stratified variable | Anaesthesia | Crude-HR (95% CI) | *P* value | *P* value (interaction) | PS adjusted-HR (95% CI) | *P* value | PS matched-HR (95% CI) | *P* value |
|---|---|---|---|---|---|---|---|---|
| **Nonstratified** | Desflurane | 1.00 | 0.003 | | 1.00 | | 1.00 | |
| | Propofol | 0.11 (0.03–0.47) | | | 0.12 (0.03–0.49) | 0.004 | 0.11 (0.03–0.48) | 0.003 |
| **TNM stage** | | | | 0.926 | | | | |
| TNM: I | | (cannot converage) | | | (cannot converage) | | (cannot converage) | |
| TNM: II+III | Desflurane | 1.00 | | | 1.00 | | 1.00 | |
| **Disease progression** | Propofol | 0.13 (0.03–0.55) 1.00 | 0.005 | | 0.13 (0.03–0.57) 1.00 | 0.006 | 0.13 (0.03–0.57) 1.00 | 0.007 0.038 |
| BCR | Desflurane Propofol | 0.20 (0.05–0.91) | 0.037 | | 0.20 (0.04–0.88) | 0.033 | 0.20 (0.05–0.91) | |

HR = hazard ratio; PS = propensity score; TNM = tumour–node–metastasis; BCR: biochemical recurrence.

The perioperative stress induced by surgery leads to metabolic and neuroendocrine changes that result in significant depression of cell mediated immunity. [3] This can result in the emergence of micro-seed tumor cells during surgery, which can avoid host immune surveillance, and eventually result in tumor recurrence or metastasis. [3] The preclinical data showed that different anesthetic techniques or anesthetics might influence the immune system in different ways [4–9] and affect risks of cancer recurrence or metastasis or the cancer patient's survival. [6,8–11] Postoperative biochemical recurrence has impacts on patient prognosis following prostate cancer surgery; thus, studies on prostate cancer have focused on searching paths to ameliorate overall patient survival via reducing them. [1,27] In addition, Biki et al. [1] reported that epidural anesthesia/analgesia was associated with less risk of biochemical cancer recurrence in patients undergoing radical prostatectomy. However, Wuethrich et al. [28] showed that epidural analgesia did not reduce the risk of biochemical cancer recurrence or improve survival following open radical prostatectomy, and we did not routinely perform regional anesthesia/analgesia in RARP in our hospital.

Data from human prostate cancer cell lines supported the influence of propofol on prostate cancer cell growth and survival via reducing hypoxia-inducible factor (HIF)-1α expression. [29–30] Qian et al. [29] reported that propofol might inhibit prostate cancer progression and metastasis via decreasing HIF-1α expression and reversing hypoxia-induced epithelial-mesenchymal transition by suppressing HIF-1α. Huang et al. [30] showed that propofol reduced HIF-1α expression in prostate cancer cells. Upregulation of HIF was associated with a poor prognosis in colorectal cancers study. [31] In addition, HIF-1α was overexpressed in pancreatic cancer, [32] and a knockout of HIF-1α suppressed the metastasis of pancreatic cancer. [33] By contrast, previous researches showed that isoflurane had deleterious effects on the upregulation of HIF and stimulated angiogenesis in prostate and renal cancer cells. [30,34] Taken together, these limited reports suggested that the administration of isoflurane [3,30] or sevoflurane [3,11,12] may stimulate tumor cell growth, whereas propofol had a beneficial effect by suppressing tumor cell growth. [3,11,12] However, to our knowledge, the mechanism by which desflurane anesthesia influences the recurrence or metastasis of prostate cancer remains unknown. In addition, Tatsumi et al. [35] revealed that propofol suppressed nuclear androgen receptor protein levels, and inhibited androgen receptor transcriptional activity and proliferation in prostate cancer cells. These findings suggest that propofol may be a useful drug for treating prostate cancer, though further clinical studies are needed.

In this study, we showed that all-cause mortality was 4.1% after RPAP. In the matched groups, all-cause mortality and cancer-specific mortality were 4.4% and 0.8%, respectively. The result was similar with a previous study reporting that the all-cause mortality was 3.9%. [36] However, the cancer-specific mortality was lower than previous reports (1.7% and 4.3%). [37,38] This might be due to the higher surgical volume at our center and higher level of experience because we excluded patients from the learning curve period of the first two years of surgery performed by our teams. [39,40]

There were some limitations in the current study. First, it was retrospective and the 631 patients were not randomly allocated. Patient characteristics such as time since the earliest included patient and calendar period differed significantly between the groups, and we conducted PS matching to deal with this issue, and PS matching may minimize confounding in this observational study. [41] However, the small groups for propensity matching may influence the rigorousness of the statistical significance in our study. Fortunately, regardless the analytic approaches were applied, the point estimates and significances of relative risk of propofol versus desflurane were similar. Further prospective multi-center study is warranted. Second, different VAs may have different effects on prostate cancer. We analyzed only desflurane anesthesia because it is the most frequently used VA in our hospital. Third, we did not

routinely use nonsteroidal anti-inflammatory drugs during prostate cancer surgery in our hospital, because of the risk of life-threatening complications such as peptic ulceration [42], and nonsteroidal anti-inflammatory drugs were not associated with prostate cancer survival. [43] Fourth, intraoperative opioids administration was associated with an increased risk of cancer recurrence after prostate cancer surgery, [44] but information about opioid use, especially for postoperative pain control, was incomplete in the medical records used in our study. However, we presumed there was no significant difference between the two anesthetic techniques in the use of intraoperative opioids in the patients included in our current study as previous laparoscopic surgery research. [45] Fifth, previous studies reported that high hospital or surgeon volumes were significantly associated with positive patient outcomes in prostate cancer surgery. [46–47] We conducted the surgeon analysis (n = 9) and the result showed that, in our high volume surgery center, the postoperative outcome was not affected by which surgeon performed the procedure (Table 2). Finally, the anesthesiologists chose the type of anesthesia, which may have been subject to original selection bias between propofol and desflurane anesthesia. However, Jaeger et al. [48] concluded that anesthesiologist volumes were not associated with postoperative mortality or long-term survival after radical cystectomy for bladder cancer. Therefore, the postoperative outcome might not affect by anesthesiologists in RAPR in our high anesthesiologist volume hospital.

In conclusion, propofol anesthesia in RARP was associated with improved survival and lower risk of postoperative BCR compared with desflurane. Further investigations are needed to inspect the influences of propofol anesthesia on patient outcomes of prostate cancer surgery.

## Acknowledgments

The authors thank the Cancer Registry Group of Tri-Service General Hospital for the clinical data support.

## Author Contributions

**Conceptualization:** Hou-Chuan Lai, Zhi-Fu Wu.

**Data curation:** Meei-Shyuan Lee, Kuen-Tze Lin, Yi-Hsuan Huang, Jen-Yin Chen, Yao-Tsung Lin, Kuo-Chuan Hung.

**Formal analysis:** Meei-Shyuan Lee, Kuen-Tze Lin, Yi-Hsuan Huang, Jen-Yin Chen, Yao-Tsung Lin, Kuo-Chuan Hung.

**Investigation:** Hou-Chuan Lai.

**Methodology:** Hou-Chuan Lai, Meei-Shyuan Lee, Zhi-Fu Wu.

**Software:** Meei-Shyuan Lee.

**Supervision:** Zhi-Fu Wu.

**Writing – original draft:** Hou-Chuan Lai, Meei-Shyuan Lee.

**Writing – review & editing:** Zhi-Fu Wu.

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
