## [Decision Letter · Decision Letter 0]

12 Feb 2020

PONE-D-19-34968

Propofol-Based Total Intravenous Anesthesia Is Associated with Better Survival Than Desflurane Anesthesia in Robot-Assisted Radical Prostatectomy

PLOS ONE

Dear Dr.Zhi-Fu Wu

Thank you for submitting your manuscript to PLOS ONE. After careful consideration, we feel that it has merit but does not fully meet PLOS ONE’s publication criteria as it currently stands. Therefore, we invite you to submit a revised version of the manuscript that addresses the points raised during the review process.

ACADEMIC EDITOR: I would appreciate if you reply carefully to the reviewers' comments. 

We would appreciate receiving your revised manuscript by Mar 28 2020 11:59PM. To enhance the reproducibility of your results, we recommend that if applicable you deposit your laboratory protocols in protocols.io, where a protocol can be assigned its own identifier (DOI) such that it can be cited independently in the future. For instructions see: http://journals.plos.org/plosone/s/submission-guidelines#loc-laboratory-protocols

We look forward to receiving your revised manuscript.

Kind regards,

Ehab Farag, MD FRCA FASA

Academic Editor

PLOS ONE

Journal Requirements:

Additional Editor Comments (if provided):

Reviewers' comments:

Reviewer's Responses to Questions

**Comments to the Author**

1. Is the manuscript technically sound, and do the data support the conclusions?

Reviewer #1: Yes

Reviewer #2: Partly

2. Has the statistical analysis been performed appropriately and rigorously? 

Reviewer #1: Yes

Reviewer #2: No

3. Have the authors made all data underlying the findings in their manuscript fully available?

Reviewer #1: Yes

Reviewer #2: Yes

4. Is the manuscript presented in an intelligible fashion and written in standard English?

Reviewer #1: Yes

Reviewer #2: No

5. Review Comments to the Author

Reviewer #1: This study investigates the effect of anesthetic technique on survival in patients undergoing prostatectomy for prostate cancer, retrospectively reviewing data over a 10 year period.

Overall comments:

1. The authors were diligent with comparison between subtypes, including TNM classification and histology. Unfortunately this meant small groups for propensity matching which influences the rigorousness of their statistical significance.

2. There was wide variability in the HR for overall survivability, but results were still meaningful.

3. Their overall conclusion that results were significant but require further investigation is appropriate.

Recommendation: Accept with minor edits.

Copyedits:

1. Abstract conclusion: Remove "Besides", substitute "In addition,".

2. Introduction, page 9, line 6. Remove "Besides,"

3. Introduction, page 9, Line 9-10, change to: Unfortunately, recurrence of prostate cancer in patients after surgery results in...

4. Introduction, page 9, line 10-11: change to "Surgical intervention itself may result in neuroendocrine and metabolic changes which may lead to impaired cell-mediated immunity and activation of circulating tumor cell implantation."

5. Introduction, page 9, line 15: Change to "Research has shown that volatile anesthetics are pro-inflammatory..."

6. Introduction, page 6, line 6: change to "We hypothesized that patients under desflurane anesthesia may have poorer outcomes than patients under propofol..."

7. In Methods section, page 11, same units should be used for flow: 300 mL/min

8. Page 13, please change to: "Because significant interactions with the two anesthetic techniques (propofol or desflurane) were found, we also performed subgroup analyses for TNM stage as well as postoperative BCR." Please add detail to statement: Because significant interactions with the two anesthetic techniques (propofol or desflurane) were found,

9. page 14, end of page: "This finding did not change substantially in the multivariable analyses after adjustment for those significant variables in the univariable analyses and 9 surgeons (HR, 0.12; 95% CI, 0.03–0.54; P = 0.006)". Do you mean after adjustment for significant variables including the nine surgeons?

10. page 15, "After the multivariable analysis, a higher preoperative PSA level was another variable that was identified thatsignificantly increased the mortality risk.

11. page 15, end of page: In summary, patients with desflurane anesthesia had more significant disease progression (such as postoperative BCR) than those with propofol anesthesia.

12. Page 16, in discussion: "The major findings in the present study are that propofol anesthesia in RARP improves survival and reduces postoperative BCR compared with desflurane anesthesia.The results were consistent with our previous reports demonstrating that propofol anesthesia..."

13. Page 16, discussion: "The result was consistent with Coughlin et al., who reported that..."

14. Page 16, discussion: "However, previous data indicated that considerable number of men had

prostate cancer despite being within normal PSA range. Importantly, many of these patients were later found to have high grade histology."

15. Page 16, discussion, end of page: "The perioperative stress induced by surgery leads to metabolic and neuroendocrine changes that result in significant depression of cell mediated immunity. This can result in the emergence of micro-seed tumor cells during surgery, which can avoid..."

16. Page 17, discussion:" In addition, HIF-1 was overexpressed in pancreatic cancer,32 and a knockout of HIF-1..."

17. Page 18, discussion: " However, the cancer-specific mortality was lower than previous reports (1.7% and 4.3%). This might be due to the higher surgical volume at our center and higher level of experience because we excluded patients from the learning curve period of the first two years of surgery performed by our teams."

18. Page 19, discussion: "We conducted the surgeon analysis (n=9) and

the result showed that, in our high volume surgery center, the postoperative outcome was not affected by which surgeon performed the procedure (Table 2). Finally, the anesthesiologists chose the type of anesthesia, which may have been subject to original selection bias..."

Reviewer #2: This is a retrospective study in which the authors evaluate the effect of propofol vs desflurane anesthesia on survival. Propofol anesthesia improved overall survival in robot-assisted radical

prostatectomy compared with desflurane anesthesia. Furthermore, patients with propofol anesthesia had less postoperative biochemical recurrence.

This is an interesting study adding to literature that supports the beneficial effects of propofol on overall survival and cancer recurrence. However, I am concerned about the statistics. First of all the authors did not distinguish between mediators and confounders in their analysis, which means that thy might have over-adjusted. Second, I do not see any correction for multiple comparisons.

Specific comments

Abstract:

Methods: please specify the outcomes

Page 8: the authors use many abbreviations, which makes the manuscript hard to read

Introduction

Page 9, para 1: please change to: Unfortunately, recurrence of prostate cancer after surgery increases postoperative morbidity and mortality.

Page 9, para 1: which may impair cell-mediated…………

Page 10 para 1: Please move the discussion in regards to neuraxial anesthesia to the discussion section. It inhibits the flow of the introduction.

Page 10, para 1, hypothesis: please be more specific. What is your primary and what is the secondary hypothesis?

Page 11, para 2: “The exclusion criteria were propofol anesthesia combined with VAs or regional analgesia” I assume that regional anesthesia was excluded for the desflurane patients as well?

Page 11, last para: did you really use 100% oxygen?

Page 12, para 2: “time since earliest included patient”. What does that mean? Why did you not use “year of surgery”?

Page 13, para 1: you list many potential confounding factors. Please be careful in distinguishing whether these variables are true confounders or mediators. A confounder is associated with both the treatment as well as the outcome, while a mediator is associated with the outcome.

F.e Tumor staging is associated with the outcome (survival) but not with the treatment (propofol or desflurane anesthesia) and thus is a mediator for which you cannot adjust.

Page 13, para 2: The primary outcome was overall survival? Or was it time to death?

Page 13: the authors make several comparisons. How did you correct for multiple comparisons?

Page 14, para 3: “according to the anesthetic technique and other variables were compared individually in a” What are the other variables?

Table 1: Please use standardized difference for the matched variables (and not p-value)

Page 18, limitations: as you excluded patients with regional anesthesia your fith limitation is actually not a limitation.

Please be consistent with the tense (past or present) you use throughout the manuscript.

6. PLOS authors have the option to publish the peer review history of their article (what does this mean?). If published, this will include your full peer review and any attached files.

Reviewer #1: No

Reviewer #2: No

---

## [Author Response · Author response to Decision Letter 0]

18 Feb 2020

Please see the Response to Reviewers File.

---

## [Editor Report · Decision Letter 1]

26 Feb 2020

Propofol-Based Total Intravenous Anesthesia Is Associated with Better Survival Than Desflurane Anesthesia in Robot-Assisted Radical Prostatectomy

PONE-D-19-34968R1

Dear Dr.Zhi-Fu Wu

We are pleased to inform you that your manuscript has been judged scientifically suitable for publication and will be formally accepted for publication once it complies with all outstanding technical requirements.

With kind regards,

Ehab Farag, MD FRCA FASA

Academic Editor

PLOS ONE
---

## [Editor Report · Acceptance letter]

28 Feb 2020

PONE-D-19-34968R1 

Propofol-Based Total Intravenous Anesthesia Is Associated with Better Survival Than Desflurane Anesthesia in Robot-Assisted Radical Prostatectomy 

Dear Dr. Wu:

I am pleased to inform you that your manuscript has been deemed suitable for publication in PLOS ONE. Congratulations! Your manuscript is now with our production department. 

With kind regards,

on behalf of

Dr. Ehab Farag 

Academic Editor

PLOS ONE